# PopPK and PBPK Models Guide Meropenem Dosing in Critically Ill Children with Augmented Renal Clearance

**DOI:** 10.3390/pharmaceutics17121544

**Published:** 2025-11-29

**Authors:** Yao Liu, Hua He, Sa-Sa Zhang, Jia Zhou, Jin-Wei Zhu, Jin Xu, Hong-Jun Miao, Ji-Hui Chen, Kun Hao

**Affiliations:** 1State Key Laboratory of Natural Medicine, Jiangsu Province Key Laboratory of Drug Metabolism and Pharmacokinetics, China Pharmaceutical University, Nanjing 210009, China; liuyao_nch@njmu.edu.cn (Y.L.);; 2Department of Pharmacy, Children’s Hospital of Nanjing Medical University, Nanjing 210008, China; 3Department of Pharmacology, School of Pharmacy, China Pharmaceutical University, Nanjing 210009, China; 4Department of Clinical Pharmacy, Xinhua Hospital Affiliated to Shanghai Jiao Tong University School of Medicine, Shanghai 200082, China; 5Department of Emergency, Children’s Hospital of Nanjing Medical University, Nanjing 210008, China

**Keywords:** pediatric intensive care, model-informed precision dosing, population pharmacokinetic model, physiologically based pharmacokinetic model, meropenem, renal function

## Abstract

**Background:** Meropenem (MEM) is frequently prescribed for the empirical management of severe infections in the pediatric intensive care unit (PICU). Critically ill children exhibit substantial pharmacokinetic (PK) variability, and current dosing strategies remain inadequately evaluated, particularly in neonates, infants, and those with altered renal function. **Methods:** This study employed a dual modeling approach integrating population pharmacokinetic (PopPK) and physiologically based pharmacokinetic (PBPK) methodologies. Clinical data from two PICUs were utilized for PopPK model development and PBPK model evaluation. Both models were rigorously assessed using goodness-of-fit plots and prediction-based metrics. Monte Carlo simulations were subsequently conducted to calculate the probability of target attainment (PTA) for multiple dosing regimens across MICs of 0.25–16 mg/L. The pharmacodynamic target (PDT) was defined as maintaining unbound plasma concentrations above the MIC for 100% of the dosing interval (100% ƒT _> MIC_), and dosing regimens were considered acceptable if the PTA exceeded 90% for efficacy while avoiding potential toxicity (Css ≥ 50 mg/L). Results: A total of 202 MEM plasma concentrations from 101 pediatric patients were analyzed. Marked inter-individual variability in MEM pharmacokinetics and pharmacodynamics was observed. Augmented renal clearance (ARC) was frequently identified in PICU patients. We simultaneously developed a two-compartment population pharmacokinetic model incorporating body weight and estimated glomerular filtration rate, and a whole-body physiologically based pharmacokinetic model scaled from adults with adjustments for transporter ontogeny and renal function. The PopPK model, by incorporating interindividual variability on clearance and volume of distribution, captured a wider range of drug exposures and demonstrated superior predictive performance, particularly in subgroups with high eGFR. The PBPK model showed higher precision in the low eGFR subgroup but slightly lower overall predictive accuracy. Both models identified ARC as a key driver of subtherapeutic exposure. Standard regimens were insufficient for preterm neonates when the MIC was ≥4 mg/L, and even the maximum label-recommended dose failed to achieve the pharmacodynamic target for infants older than 1 month when the MIC was ≥2 mg/L. **Conclusions:** Both PBPK and PopPK frameworks reliably predicted MEM pharmacokinetics in critically ill pediatric patients, with complementary strengths across renal function strata. Model-informed simulations highlighted the inadequacy of standard dosing under conditions of ARC or elevated MIC, supporting individualized, precision-guided dosing strategies based on age, eGFR, and pathogen MIC.

## 1. Background

Severe infections remain a major contributor to in-hospital mortality among patients in the intensive care unit (ICU) [1]. Meropenem (MEM), a broad-spectrum carbapenem, is widely used as empiric therapy in pediatric and neonatal ICUs; however, its use in infants younger than 3 months and in children with renal impairment is frequently off-label [2,3]. Critically ill children exhibit pronounced inter- and intra-patient pharmacokinetic (PK) variability, making label regimens derived from relatively healthy cohorts potentially suboptimal in the pediatric intensive care unit (PICU) setting [4,5]. 

Model-informed precision dosing has increasingly been recognized as a valuable strategy for optimizing antibiotic therapy in critically ill populations. Although conventional population pharmacokinetic (PopPK) modeling has been employed to characterize MEM PK behavior across various adult subgroups, including obese, elderly, and renally impaired patients [6,7,8,9], these studies are often limited by small sample sizes and homogeneous cohorts [10]. Consequently, the generalizability of such models to heterogenous, critically ill pediatric populations remains uncertain, and existing model-based dosing recommendations remain inconsistent across studies [11,12,13].

Physiologically based pharmacokinetic (PBPK) modeling provides a bottom-up framework to mechanistically describe drug disposition by linking tissue/organ compartments via blood flows, tissue partitioning, and metabolism parameters [14]. For pediatrics, PBPK enables explicit incorporation of ontogeny, including age-dependent changes in organ size and blood flow, plasma protein binding, and renal processes, so that adult knowledge can be scaled to neonates, infants, and older children [14,15]. Compared with data-driven, top-down PopPK models, PBPK emphasizes biological plausibility and facilitates extrapolation across age groups and physiology states; as such, the two approaches are complementary rather than competing.

Guided by these principles, we created an adult MEM PBPK model from physicochemical and in vitro and clinical knowledge of disposition pathways and subsequently scaled it to critically ill pediatric patients. In parallel, we constructed a pediatric PopPK model from therapeutic drug monitoring (TDM) data to quantify between-subject variability and covariate effects. Using both frameworks, we evaluated steady-state exposure and Pharmacokinetic/Pharmacodynamic (PK/PD) target attainment across renal-function strata and minimum inhibitory concentration (MIC) ranges to inform model-guided dose optimization for PICU patients.

## 2. Methods

### 2.1. Data Source

This retrospective observational study was conducted in two PICUs in the Xinhua Hospital, affiliated with the Shanghai Jiao Tong University School of Medicine and Children’s Hospital of Nanjing Medical University, between January 2020 and December 2023. Age, body weight (BW), height (HT), white blood count (WBC), Neutrophil Percentage (Neut%), alanine transferase (ALT), aspartate transaminase (AST), serum creatinine (Scr), albumin (ALB), total protein (TP), and daily dosage were retrospectively collected. The estimated glomerular filtration rate (eGFR) was calculated using the following Schwartz equation (Equation (1)):(1)$$eGFR=\;\frac{k\times \;H}{Scr}$$
where k = 0.33 for the preterm population [16], k = 0.45 for full-term population [16], k = 0.41 for other pediatric population [17], H–height in cm, Scr–serum creatinine in mg/dL, and eGFR is expressed as mL/min/1.73 m^2^.

MEM PK profiles of healthy volunteers were (Appendix A) obtained from the published literature [18,19] and digitized using GetData^®^ Graph Digitizer (Version 2.25). MEM PK profiles of critically ill patients were sourced from the PROMESSE study (Appendix A) [19].

### 2.2. Pop-PK Model Development and Evaluation Method

The Pop-PK Model of MEM in the adult and pediatric critically ill population was performed using the Stochastic Approximation Expectation-Maximization (SAEM) algorithm in Monolix^®^ (Lixoft, France, V2023). Data processing and visualization were carried out using R^®^ software (Version 4.0.5).

We assessed one- and two-compartment models to determine the optimal structural PK model. Inter-individual variability (IIV) in typical population parameter estimates was modeled using a log-normally distribution. Residual unexplained variability was assessed using constant, proportional, and combined error models. After establishing the structural model, we evaluated demographic, clinical, and laboratory variables for potential inclusion as covariates in the PopPK model. Candidate continuous covariates included BW, HT, WBC, Neut%, ALT, AST, Scr, ALB, TP, and eGFR. Categorical covariates tested were gender and preterm status. Covariate selection was guided by biological plausibility and implemented via stepwise regression using the −2 × log-likelihood as objective function value (OFV). Correlations between η parameters were incorporated when the estimated correlation coefficient exceeded 0.30 and its inclusion produced a decrease in OFV (ΔOFV ≥ 3.84 for 1 df) with stable relative standard errors (RSEs). The final model retained covariate effects that could be estimated with a reasonable RSE, ensuring robust parameter estimates.

Model evaluation included visual inspections of goodness-of-fit (GOF) plots and was internally validated using the bootstrap method. A total of 1000 replicate datasets were generated through random sampling with replacement from the original dataset, and the 95% confidence intervals for the parameter’s estimates were computed.

### 2.3. PBPK Model Development and Evaluation Method

The PBPK model was developed using the software PK-Sim^®^ (Version 11.0; Bayer Technology Services, Leverkusen, Germany). A healthy adult model was first constructed based on MEM physicochemical properties and in vitro parameters, as shown in Appendix A. The disposition of MEM was modeled via glomerular filtration, biliary clearance, renal secretion, and DPEP1-mediated metabolism [18,20,21]. The maximum transport rate (V_max_) of OAT3-mediated uptake and a hypothetical transporter-mediated efflux which constitute the tubular secretion process for MEM were optimized using data from previous study [18]. DPEP1-mediated metabolism was assumed as first-order clearance [20] and optimized using in vivo non-renal clearance values observed in adults [18,22,23,24]. Following its validation against non-compartmental analysis (NCA) data from healthy adults, the model was adapted for critically ill patients. Adaptation involved incorporating patient-specific physiological parameters informed by local sensitivity analyses [19]. The patient adult model was further validated by generating virtual patient populations and compared with the published data [19]. Assuming similar MEM disposition between adults and children, the pediatric patient model was developed by scaling the adult patient model. The algorithms and default albumin ontogeny profiles within PK-Sim^®^ were utilized to incorporate age-dependent developmental changes. OAT3 ontogeny was adopted from reported data [25], while the hypothetical efflux transporter expression was modeled via an ontogeny function for tubular secretion [11,26]. The non-renal clearance was scaled according to the organ maturation [11]. In PK-Sim^®^, age-dependent changes in GFR have been implemented [27], defined as GFR_specific_ (mL/min/100 g, Equation (2)).GFR = GFR_specific_ × kidney weight (2)

The GFR_aj_ was defined as adjusted GFR_specific_ based on the eGFR level of real patients.eGFR = GFR_aj_ × kidney weight (3)

Predictive performance criteria (PPC) and average fold error (AFE) were used to verify the accuracy of the model’s performance. The PPC_2×_ was defined as the percentage of predicted concentrations that are within 0.5- to 2-fold of the actual observed concentrations. The performance of the model was considered acceptable if the AFEs were between 0.5 and 2 and PPC_2×_ exceeded 80% [11].

### 2.4. Evaluation of Recommended Dosing Regimens with PopPK and PBPK Models

According to the prescribing information, Simulations were conducted for six clinical scenarios: four scenarios for infants under three months of age (preterm and PNA < 2 weeks, preterm and PNA ≥ 2 weeks, term and PNA < 2 weeks, term and PNA ≥ 2 weeks), one for infant aged 1–3 months, and one for children aged three months above. Each group was further stratified into augmented renal clearance (ARC) and non-ARC subgroups. The classification of ARC versus non-ARC subgroups was determined according to developmental physiological criteria, with ARC defined as eGFR values exceeding one standard deviation above the mean (Mean + 1 SD) of the age-specific physiological reference range for healthy children (Appendix A). The evaluated dosing regimens included 20 mg/kg every 8 h or every 12 h for neonates, and 20 mg/kg every 8 h for older infants and children, administered as 60 min intravenous infusions. Virtual populations were generated using physiological parameters from our real-world data. In the PopPK model, 1000 virtual subjects per scenario were simulated based on median eGFR and BW. In the PBPK model, virtual patient populations of 1000 subjects per scenario were generated using group-average demographic information (including age, weight, eGFR) in PK-Sim^®^.

Pathogen MIC values ranging from 0.25 to 16 mg/L were considered, based on distributions for *Pseudomonas* spp. and *Acinetobacter* spp. sourced from the European Committee on Antimicrobial Susceptibility Testing (EUCAST) database [28]. For carbapenem antibiotics, the pharmacodynamic target (PDT) was defined as maintaining unbound plasma concentrations above the MIC for 100% of the dosing interval (100% ƒT _> MIC_). The probability of target attainment (PTA) was then calculated for each dosing regimen using the final PopPK and PBPK models, and regimens were considered acceptable if PTA exceeded 90% [5,29,30]. A plasma steady-state concentration (Css) ≥ 50 mg/L was set as the toxicity threshold [31].

## 3. Results

### 3.1. Pediatric Patients Demographic Information

A total of 101 pediatric patients were included in the study with demographic, clinical, and TDM data analyzed. A total of 202 measured MEM concentrations were included, of which 14 samples were below the detection limit. Neonates accounted for approximately 29% of the cohort, and infants younger than 3 months comprised 47.5%. Baseline characteristics are summarized in Table 1. The most frequent infection-related diagnoses were sepsis/sepsis shock (77.2%), pneumonia (57.4%) and bacterial meningitis (31.7%). Each group’s levels were higher than the data of healthy pediatrics. The patients received various MEM dosing regimens, with daily doses ranging from 22 to 157 mg/kg, and dosing intervals from every 24 h to every 6 h.

### 3.2. PopPK Model Development and Evaluation

A two-compartment model with linear elimination effectively captured the MEM concentration profile, incorporating log-normal IIV in clearance (CL) and central volume of distribution (V1). Residual variability was effectively captured by a proportional error model. Following covariate screening by stepwise regression, BW and eGFR were identified as significant covariates in the final PopPK model. Specifically, CL was determined by both eGFR and BW, while the V1 and intercompartmental clearance (Q) were significantly influenced by BW. Consequently, the final covariate model retained BW and eGFR on CL, and BW on V1. Equation (4) in the final PK model characterized the CL:$${CL}_{i}=\theta_{CL}\times exp\left(\beta_{CL,BW}\times \frac{{BW}_{i}}{BWmed}\right)\times exp\left(\beta_{CL,eGFR}\times {eGFR}_{i}\right)\times exp\left(\eta CL,i\right)\;$$
where $\theta_{CL}$ is the typical value of endogenous clearance, $\beta_{CL,BW}$ represents the exponential factor of BW on CL, and $\beta_{CL,eGFR}$ represents the exponential factor of eGFR on CL. ηCL is the IIV of $\theta_{CL}$.

V1 was modeled as Equation (5):$${V1}_{i}=\theta_{V1}\times exp\left(\beta_{V1,BW}\times \frac{{BW}_{i}}{BWmed}\right)\times exp\left(\eta V1,i\right)$$
where $\theta_{V1}$ is the typical value of V1, and $\beta_{CL,BW}$ represents exponential factor of BW on V1. ηV1 is the IIV of $\theta_{V1}$.

Q was modeled as Equation (6):$$Q_{i}=\theta_{Q}\times exp\left(\beta_{Q,BW}\times \frac{{BW}_{i}}{BWmed}\right)$$

These parameters were accurately estimated, and their precision was confirmed by the non-parametric bootstrap analysis, detailed in Table 2.

The GOF plots of the final model are illustrated in Figure 1. Clearly, the scatterplots depicting observed concentrations against both population and individual predicted concentrations showed a consistent alignment with the line of identity. Additionally, Individual Weighted Residuals (IWRES) and Normalized Prediction Distribution Errors (NPDE) exhibited symmetric distribution around zero across the entire population predictions and time after dose range. These diagnostic plots affirm the robustness and predictive accuracy of our final model.

### 3.3. PBPK Model Development and Evaluation

#### 3.3.1. Adult PBPK Model Evaluation

The adult PBPK model adequately characterized the MEM plasma PK following 500 mg intravenous infusions. Observed MEM concentrations ± standard deviation of published adult data [18] were in good agreement with the 90% prediction interval of the simulated results. Simulated AUC and CL in healthy adult populations were within a 72%~114% range of the published data, as displayed in Appendix A. A similar observation was found (Figure 2) in the adult patient PBPK model. Simulated AUC and CL in the adult patient population were also within 75%~150% range of the published values (Table 3). In the single parameter sensitivity analysis (Appendix A), the variation in OAT3 V_max_ and GFR level significantly impacted the change in AUC.

#### 3.3.2. Pediatric PBPK Model Evaluation

Observed versus model predicted plasma concentrations of MEM in PICU patients are provided in Figure 3A, demonstrating good agreement across different dosing regimens. Overall, 85% of the predicted MEM plasma concentrations were between 50% and 200% of observed data. The AFE of the pediatric model, including all samples, was 0.99. The AFEs for the preterm group, term group, age < 3 months group, and age ≥ 3 months group were 0.93, 1.03, 1.06, and 0.98. The overall AFE indicated a good model performance in pediatric patient populations of different age groups. As seen in Figure 3C, the pred/obs increased with higher levels of eGFR_aj_/GFR_specific_, suggesting that deviations of eGFR from baseline are associated with greater prediction bias. In contrast, age, BW, gender, diagnosed with sepsis, plasma Alb, and CRP level did not show a similar impact (Appendix A).

### 3.4. Performance Comparing Between PBPK and PopPK Model

Both models demonstrated adequate calibration (Table 4). The PopPK model exhibited lower systematic bias (MPE: 13.83 vs. 14.80) but marginally higher dispersion (RMSE: 4.37 vs. 4.09) compared to PBPK model, along with superior twofold prediction accuracy (PPC_2×_: 94.8% vs. 87.7%). When the pediatric patients were stratified by eGFR tertiles, we have found that in the low eGFR group, PBPK showed superior precision (RMSE 3.83 vs. 5.69, *p* = 0.032) but lower PPC_2×_ (96.2% vs. 98.1%), suggesting a trade-off between precision and robustness. While in the moderate and high eGFR groups, PopPK demonstrated unequivocal superiority (moderate: RMSE 4.41 vs. 4.99; high: RMSE 2.28 vs. 3.21).

The estimated Glomerular Filtration Rate (eGFR) was grouped based on the tertiles of real world data, with the specific grouping criteria as follows: eGFR (Low): <102.48 mL/min/1.73 m^2^; eGFR (moderate): 102.48~172.8 mL/min/1.73 m^2^; eGFR (high): >172.8 mL/min/1.73 m^2^.

### 3.5. Dose Regimen Evaluation

In different scenarios (neonates with post-natal age < 14 days receiving q12h regimens; term neonates ≥ 14 days on q8h; infants 1–3 months; and >3 months on q8h, all dosed at 20 mg/kg/dose over 60 min), both models predicted similar central tendencies for AUC_ss_, C_max,ss_, and C_trough,ss_ (Figure 4 and Appendix A). However, PopPK generated wider prediction intervals (particularly for C_trough,ss_), indicating its ability to capture greater between-subject variability through log-normal IIV on CL and V_1_ with covariance. In contrast, the PBPK model yielded narrower distributions due to physiology-driven parameterization.

**Table 5 pharmaceutics-17-01544-t005:** Age and renal function based Recommended Dose Regimen.

eGFR/GFR (Percentage of Expected GFR Level)	MIC (mg/L)	Recommended Dose Regimen							
		<3 Months				≥3 Months			
		Dose (mg/Kg)	Infusion Duration (h)	Frequency	Daily Dose (mg/Kg)	Dose (mg/Kg)	Infusion Duration (h)	Frequency	Daily Dose
50% Renal impairment	≤1	10	1	Q8h	30	10	1	Q8h	30
	(1, 4]	10	3	Q8h	30	20	3	Q8h	60
	(4, 8]	10	3	Q8h	30	40	3	Q8h	120
	(8, 16]	20	1	Q8h	60	Not Recommended			
100% No ARC	≤1	10	1	Q8h	30	10	3	Q8h	30
	(1, 4]	10	3	Q8h	30	40	1	Q8h	120
	(4, 8]	20	3	Q8h	60	40	3	Q8h	120
	(8, 16]	40	3	Q8h	120	Not Recommended			
150% Moderate ARC	≤1	10	1	Q8h	30	10	3	Q8h	30
	(1, 4]	20	1	Q8h	60	40	3	Q8h	120
	(4, 8]	20	3	Q8h	60	60	1	Q8h	180
	(8, 16]	40	3	Q8h	120	Not Recommended			
200% Severe ARC	≤1	10	1	Q8h	30	10	3	Q8h	30
	(1, 4]	40	1	Q8h	120	40	3	Q8h	120
	(4, 8]	40	3	Q8h	120	Not Recommended			
	(8, 16]	Not Recommended				Not Recommended			

## 4. Discussion

MEM is commonly employed for empiric anti-infective therapy in pediatric and neonatal ICUs due to its potent antibacterial spectrum and favorable safety profile. In this dual-center cohort of critically ill pediatric patients, we developed two complementary modeling frameworks for MEM: a data-driven PopPK model with parameters estimated from pediatric TDM data and a PBPK model qualified in healthy adults and extrapolated to pediatric patients with ontogeny and eGFR-individualized renal clearance.

Our cohort exhibited a pronounced rightward shift in renal function. Among patients older than 3 months, median eGFRs (175.8 for 3 months–<2 years; 164.7 for 2–<6 years; 177.9 for 6–<15 years; mL/min/1.73 m^2^) were clearly above age-specific healthy references, consistent with widespread ARC in the PICU setting [33]. Although serum creatinine-based eGFR estimation is challenging in infants under one month due to maternal creatinine influence and fluid shifts [34], both term and preterm neonatal subgroups exhibited hyperfiltration relative to healthy references. Term neonates had a median eGFR of 88 mL/min/1.73 m^2^, substantially exceeding the reference value of 59 mL/min/1.73 m^2^ at 4 weeks, while the upper quartile in preterm neonates reached 133 mL/min/1.73 m^2^. Taken together, label doses extrapolated from healthy or mildly ill pediatric populations are likely to underestimate exposure needs for renally cleared agents in PICU neonates and infants [35].

Given this pronounced variability, we compared two models’ predictive performance and how each modeling framework captured the impact of eGFR on MEM disposition. It is important to recognize a key distinction in model development: unlike the PopPK model which was developed and validated using sparse sampling data directly obtained from the PICU patients, the PBPK model was constructed from healthy adult data and subsequently extrapolated to the pediatric population. Both models demonstrated acceptable predictive performance. The PopPK model exhibited marginally better fold accuracy and bias, whereas RMSE was comparable between the two approaches. These results indicate that the PBPK model possesses strong generalizability, making it particularly suitable for drug evaluation in pediatric populations where dense sampling is impractical. Discrepancies between the two approaches became most apparent when stratifying by renal function. The PopPK demonstrated slightly lower bias and higher AFE overall and captured heterogeneity more fully at mid/high eGFR. In contrast, the PBPK model showed superior precision at low eGFR levels, consistent with previous findings showing that incorporation of developmental physiology improves prediction in patients with renal dysfunction [36]. These differences suggest that in patients with ARC, accelerated MEM elimination may involve mechanisms beyond glomerular filtration alone. Despite relying on adult-to-pediatric extrapolation and eGFR-based individualization, the PBPK model achieved robust overall calibration and maintained high predictive precision. Consequently, both models yielded consistent conclusions in steady-state exposure and PTA simulations.

Across six commonly used clinical scenarios, both models produced similar central estimates of steady-state exposure but converged in identifying a critical limitation: standard regimens are frequently insufficient, and shortfalls worsen with higher MIC and enhanced renal function (high eGFR/ARC). In line with this, the PopPK study by Rapp et al. reported a median eGFR of 151.5 mL/min/1.73 m^2^ in their cohort, with widespread ARC identified as the primary driver of accelerated MEM clearance and failure to achieve target plasma concentrations [37]. When MIC is low and renal function is normal, model-based doses align with label/guideline recommendations; however, in ARC-prone patients or at higher MICs, prolonged infusion and/or increased dosing frequency are warranted. Simulations from both models revealed substantial interindividual variability in MEM pharmacokinetics and PTA, even within narrow eGFR ranges. This variability reflects dynamic changes in renal function and other treatment-related factors, such as fluid resuscitation during severe infection, which lead to substantial interindividual differences in drug exposure. These findings underscore the potential utility of TDM, particularly in unstable patients, to ensure optimal exposure. This interpretation is further supported by the external evaluation of Li et al., which demonstrated that existing MEM models perform poorly in a priori prediction, but predictive accuracy improves markedly with the integration of just one or two TDM samples [38]. Based on these findings, we advocate for a comprehensive clinical management pathway: initial therapy should employ optimized strategies such as prolonged or continuous infusion for patients with ARC or high-MIC pathogens, followed by early implementation of TDM within a validated PK model framework to enable fully individualized therapy.

Our study had several limitations that should be acknowledged. First, this dual-center retrospective design may introduce selection and information bias. Sparse TDM sampling and limited phlebotomy quality control could affect parameter identifiability and the residual error structure. Second, SCr-based eGFR is susceptible to inaccuracy in critically ill children due to non-steady-state creatinine kinetics, dilution from fluid resuscitation, and reduced creatinine production. This is especially relevant in neonates, where eGFR may substantially deviate from true glomerular filtration. Third, we did not link PK/PD target attainment to clinical outcomes such as microbiological clearance, organ recovery, length of stay, or mortality. Prospective studies are warranted to validate the clinical utility of our model-based recommendations.

## Figures and Tables

**Figure 1 pharmaceutics-17-01544-f001:**
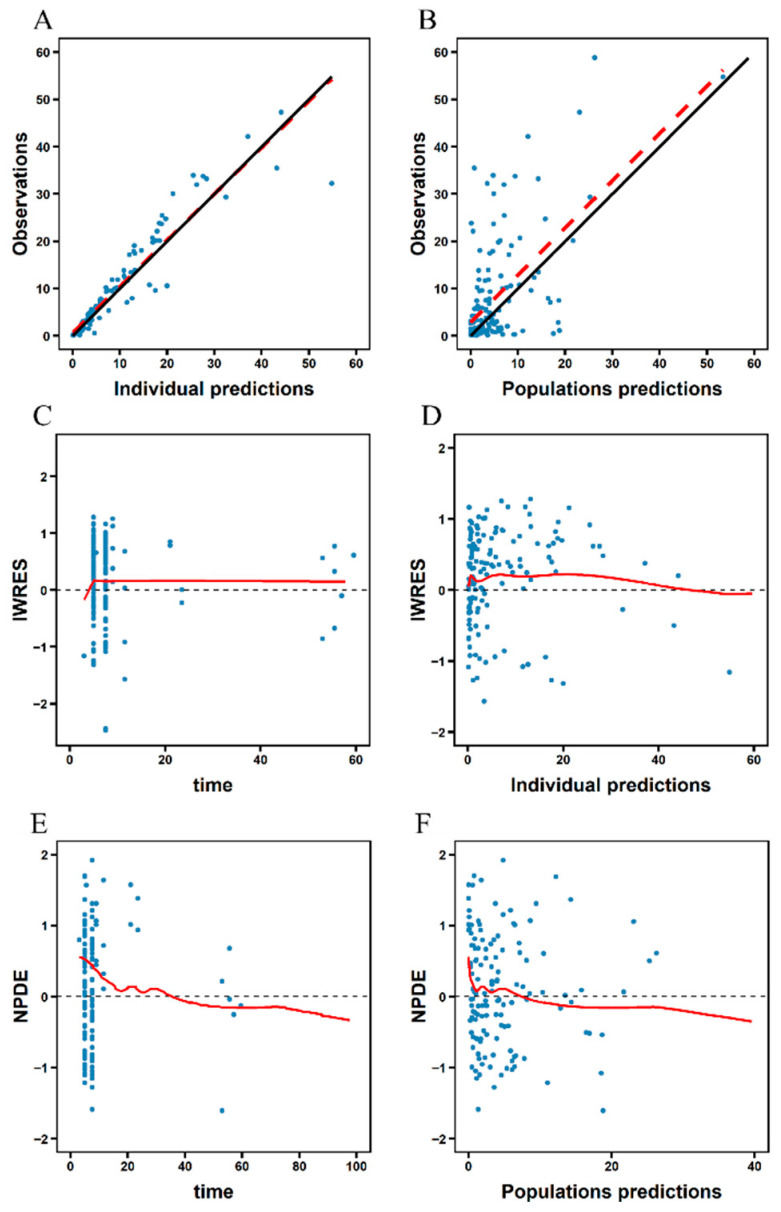
Goodness-of-fit (GOF) plots for the final PopPK model of meropenem in critically ill pediatric patients. (**A**,**B**): Population and individual predicted meropenem concentrations versus observed concentrations; (**C**,**D**): The IWRES distributions over time and individual predicted meropenem concentrations; (**E**,**F**): The NPDE distributions over time and individual predicted meropenem concentrations. The blue dots represent observed data points, and the red trend lines illustrate overall fitted trends.

**Figure 2 pharmaceutics-17-01544-f002:**
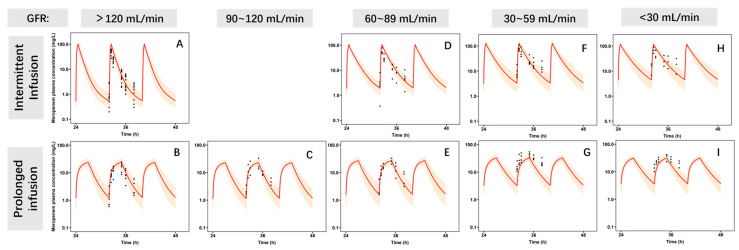
Simulation of MEM plasma concentration vs. time for virtual adult patient populations. The adult patient MEM physiologically based pharmacokinetic model was developed and evaluated using digitized published data (22). The dose regimen was 1000 mg, q8h. (**A**) IVD 0.5 h, GFR: 120 mL/min; (**B**) IVD 3 h, GFR: 120 mL/min; (**C**) IVD 3 h, GFR: 90–120 mL/min; (**D**) IVD 0.5 h, GFR60–89 mL/min; (**E**) IVD 3 h, GFR60–89 mL/min; (**F**) IVD 0.5 h, GFR30–59 mL/min; (**G**) IVD 3 h, GFR30–59 mL/min; (**H**) IVD 3 h, GFR < 30 mL/min; (**I**) IVD 0.5 h, GFR < 30 mL/min. The observed mean plasma concentration vs. time data (black circles are observed values) were compared with the 95% prediction interval of the simulated meropenem plasma concentration vs. time profile (red line with the orange shaded region). IVD, infusion time; h, hour; GFR, glomerular filtration rate.

**Figure 3 pharmaceutics-17-01544-f003:**
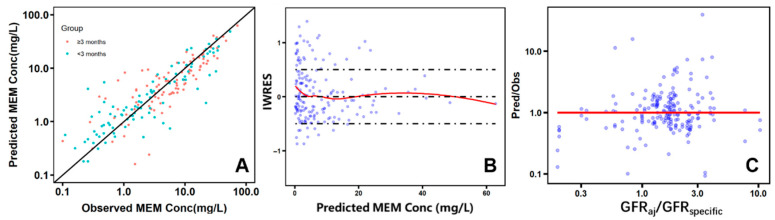
Predicted and observed meropenem plasma concentrations using PBPK modeling. (**A**): Observed concentration versus the predicted concentration (overall RMSE = 4.94, AFE = 1.08); (**B**): Individual Weighted Residuals versus the predicted concentration, blue dots represent individual residuals and the red line represents the smoothed trend. The dashed lines indicate the ±1 boundaries; (**C**): predicted concentration/observed concentration versus GFR_aj_/GFR_specific_, the red line indicates the reference value Pred/Obs = 1. MEM, meropenem; RMSE, root-mean-square error across individual concentrations; AFE, average fold error across individual concentrations; Conc, plasma concentration; IWRES, Individual Weighted Residuals.

**Figure 4 pharmaceutics-17-01544-f004:**
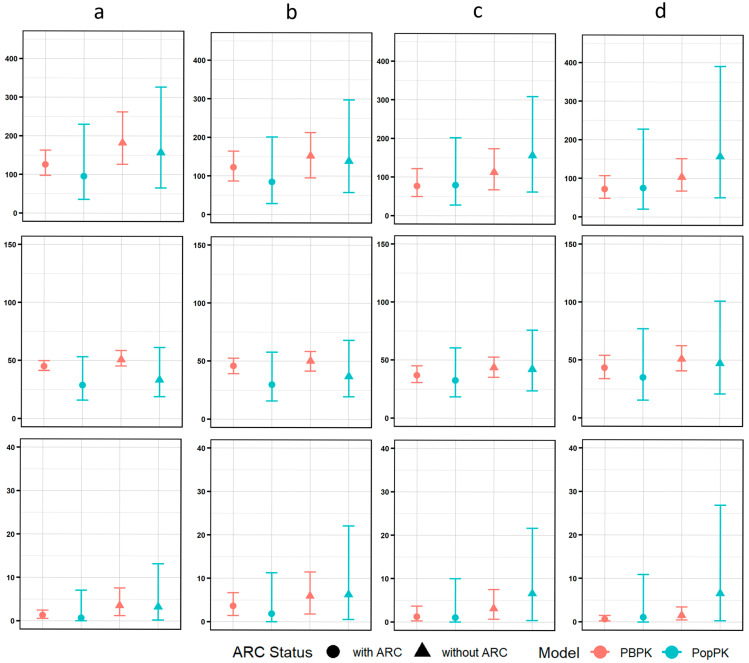
Comparison of simulated meropenem exposures parameters for the proposed doses using PopPK and PBPK approaches. The red and blue solid dots represent the mean values of PK parameters (AUC_ss_, C_max_, and C_trough_) predicted by the PBPK and PopPK models. The red and blue vertical lines indicate the 5th and 95th percentiles across the virtual pediatric populations generated by each model. Circular symbols denote children with augmented renal clearance (ARC), while triangular symbols represent children without ARC. (**a**): Term and PNA < 2 weeks, 20 mg/kg, q12h; (**b**): term and PNA ≥ 2 weeks, 20 mg/kg, q8h; (**c**): 1–3 months, 20 mg/kg, q8h; (**d**): ≥3 months, 20 mg/kg, q8h. All regimens are administered as a 60 min intravenous infusion. ARC, augment renal clearance; PNA, postnatal age; PopPK, Population Pharmacokinetic; PBPK, Physiologically Based Pharmacokinetic. The heatmap of %ƒT _> MIC_ over a wide MIC range with different dosing regimens in 4 groups patients (pretern, term, 1~3 months, above 3 months) is shown in Appendix A. %ƒT _> MIC_ above 90% was coded with red color to indicate satisfactory target attainment. A total of 20 mg q12h, which is recommend by London Health Science Center (LHSC) [32], is not suffient for preterms with severe infection when MIC is above 4 mg/L, even if infusion duration is extended to 3 h. A total of 40 mg q8h 3 h infusion, the maximum recommended dosage in the product lable, fails to achieve the PDT for patients above 1 months when MIC is above 2 mg/L. We established an individualized dosing regimen (Table 5), tailored to patient age, eGFR, and pathogen MIC, to guarantee the achievement of the PDT.

**Table 1 pharmaceutics-17-01544-t001:** Characteristics of enrolled patients.

Total	101
Age	
Total (weeks, median [IQR])	20.3 [3.2, 114.6]
Neonates (<28 d) n (%)	29 (28.7%)
Preterm (n)	13
Term (n)	16
28 d ~ < 3 m n (%)	19 (18.8%)
3 m ~ < 2 y n (%)	21 (20.8%)
2 y ~ < 6 y n (%)	20 (19.8%)
6 y ~ < 14 y n (%)	12 (11.9%)
Gender	
Female n (%)	41 (40.6%)
Male n (%)	60 (59.4%)
BW (kg g, median [IQR])	7.5 [3.4, 12.1]
BMI (kg/m^2^, mean ± SD)	15 ± 3.7
DD (mg/kg/d, mean ± SD)	125.2 ± 28.7
WBC (10^9^/L, median [IQR])	11.8 [7.9, 19.6]
NE% (%, median [IQR])	58.8 [32.8, 68.5]
ALT (U/L, median [IQR])	24.5 [14.0, 59.3]
AST (U/L, median [IQR])	38.3 [23.0, 81.0]
TP (g/dL, mean ± SD)	56.5 ± 10.28
Alb (g/dL, mean ± SD)	33.8 ± 5.35
Diagnosis related to bacterial infections	(n)
Sepsis/Sepsis shock	78
Pneumonia	58
Bacterial meningitis	32
Intra-abdominal infections	16
Skin and skin structure infections	3
Scr (μmol/L, median [IQR])	22.1 [15.5, 33.9]
eGFR (mL/min/1.73 m^2^, median [IQR])	
Total	123.4 [80.9, 180.1]
Neonates (<28 d)	56.7 [29.0, 88.4]
Preterm	33.1 [21.6, 132.6]
Term	88.0 [65.4, 109.0]
28 d ~ < 3 m	109.7 [88.4, 131.7]
3 m ~ < 2 y	175.8 [136.7, 189.5]
2 y ~ < 6 y	164.7 [122.8, 239.6]
6 y ~ < 15 y	177.9 [147.4, 183.3]

BW, body weight; BMI, body weight index; DD, daily dose; WBC, white blood count; ALT, alanine transferase; AST, aspartate transaminase; Scr, serum creatinine; Alb, albumin.

**Table 3 pharmaceutics-17-01544-t003:** Simulated and observed pharmacokinetic parameters for meropenem after intravenous infusion in adult patients.

eGFR (mL/min/1.73 m^2^)	Infusion Time (h)	Observed Mean (SD)		Simulated Mean (95%)		Ratio	
		AUC_inf_ (mg·h/L)	C_max_ (mg/L)	AUC_inf_ (mg·h/L)	C_max_ (mg/L)	AUC_inf_	C_max_
>120	0.5	70.26 (19.24)	52.91 (11.73)	120.78 (87.7–174.49)	101.85 (82.6–127.42)	0.58	0.52
	3	60.75 (20.2)	16.66 (5.77)	85.7 (60.49–122.87)	23.76 (17.89–31.47)	0.71	0.70
90–119	3	77.58 (29.56)	21.03 (10.53)	82.24 (59.54–112.28)	22.9 (17.81–28.68)	0.94	0.92
60–89	0.5	104.25 (31.48)	56.18 (12.73)	190.19 (137.69–256.98)	106.59 (95.27–118.62)	0.55	0.53
	3	94.46 (26.33)	26.42 (5.13)	100.42 (70.18–145.9)	27.22 (20.97–35.67)	0.94	0.97
30–59	0.5	194.38 (50.81)	80.74 (18.44)	246.33 (176.66–335.61)	129.5 (102.44–155.28)	0.79	0.62
	3	192.33 (45.04)	42.42 (8.32)	136.25 (96.43–184.92)	33.12 (25.91–41.69)	1.41	1.28

**Table 4 pharmaceutics-17-01544-t004:** Comparison of predictive performance between PBPK and PopPK models across renal function strata.

Group		MPE	RMSE	AFE	PPC (%)
Overall	PBPK	14.80	4.09	0.96	87.7
	Pop-PK	13.83	4.37	1.07	94.8
eGFR Low	PBPK	−5.63	3.83	0.90	96.2
	Pop-PK	5.32	5.69	1.01	98.1
eGFR Moderate	PBPK	23.14	4.99	0.96	80.8
	Pop-PK	21.99	4.41	1.11	92.3
eGFR High	PBPK	27.14	3.21	1.02	86.3
	Pop-PK	14.18	2.28	1.07	94.1

**Table 2 pharmaceutics-17-01544-t002:** Parameter estimates of the final PopPK model.

Parameter	Stochastic Approximation		Bootstrap Estimates (n = 1000)		
	Estimate	RSE (%)	Median	2.5%ile	97.5%ile
CL (L/h/kg)	0.24	11.1	0.23	0.16	0.34
V1 (L/kg)	1.53	15.8	1.64	0.9	2.79
Q (L/h/kg)	0.014	42.4	0.014	0.0002	0.038
V2 (L/kg)	6.06	23.3	6.08	9.4	18.49
$\beta_{CL,eGFR}$	0.96	9.19	0.97	0.72	1.29
$\beta_{CL,WT}$	0.43	10.3	0.42	0.33	0.65
$\beta_{V1,WT}$	0.37	19.7	0.35	0.09	0.56
$\beta_{Q,WT}$	1.54	12.1	1.62	1.04	4.27
omega_CL_	0.41	13.1	0.41	0.27	0.58
omega_V1_	0.45	21.1	0.36	0.11	0.91
Proportion error	0.34	11.6	0.34	0.24	0.41

## Data Availability

The datasets generated and/or analyzed during the current study are not publicly available due to patient privacy and ethical restrictions but are available from the corresponding author on reasonable request.

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
