# Peer review of "PopPK and PBPK Models Guide Meropenem Dosing in Critically Ill Children with Augmented Renal Clearance"

_pharmaceutics, 2025, doi:10.3390/pharmaceutics17121544_

Round 1

Reviewer 1 Report

Comments and Suggestions for Authors

In the paper entitled "opPK and PBPK Models Guide Meropenem Dosing in Critically Ill Children with Augmented Renal Clearance", the authors, following a 3 years study in pediatric patients, and considering age and physiology that could affect meropenem's pharmacokinetics, have constructed an adult meropenem physiologically based pharmacokinetic (PBPK) and then scaled it to pediatric patients. In the same time, they have developed population pharmacokinetic (PopPK) model in order to quantify between subject variability and covariate effects, proving that both PBPK and PopPK frameworks can reliably predict meropenem's pharmacokinetics in critically ill pediatric patients.

The idea of this research is not only good, but necessary. The methodology is adequate, the results are well interpretated and also, the paper is very well organised and written in English.

Only some minor changes to be made before acceptance:

  • there are some missing spaces (usually right before a reference numer);
  • there are multiple types of font used;
  • in vitro/in vitro/in-vitro (same for in vivo)...authors should be consistent;
  • I believe that "serum creatine" should be replaced with "serum creatinine".

Author Response

Dear Reviewer,

Thank you very much for your positive evaluation and valuable comments on our manuscript. We are delighted to learn that you find the idea of this research "not only good, but necessary" and that the methodology, results, organization, and English language meet a high standard. Your recognition is a great encouragement to us. We have carefully revised the manuscript according to your suggestions. The attachments include: (1) The revised manuscript showing revisions made by authors_ 3966842_R1; (2) The revised manuscript with clean version_3966842_R1_clean; (3) The revised supplementary material showing revisions made by authors_Supplementary Material_R1. All page and line numbers cited in the responses (e.g., Page*, Line*) refer to the revised manuscript file “pharmaceutics-3966842_R1.docx” with tracked changes.

Please find our point-by-point responses below:

  1. There are some missing spaces (usually right before a reference numer);

Response: We have thoroughly checked and corrected the formatting throughout the manuscript and added all missing spaces.

  1. There are multiple types of font used;

Response: We have thoroughly checked and standardized the font types used in the text. (e.g. Page5, Line157 and Line 181 and Line 183; Page7, Line256).

  1. in vitro/in vitro/in-vitro (same for in vivo)...authors should be consistent.

Response: We have standardized the format for "in vitro" to ensure consistency across the entire manuscript. (Page3, Line97; Page4, Line142).

  1. I believe that "serum creatine" should be replaced with "serum creatinine".

Response: Thank you for pointing this out. We have corrected this typo in all instances (Page5, Line 110; Page7, Line246).

Once again, we sincerely appreciate the time and effort you spent on reviewing our work and providing these meticulous suggestions, which have significantly improved the quality of our manuscript.

Reviewer 2 Report

Comments and Suggestions for Authors

Meropenem is an antibiotic from the carbapenem group that is frequently used in pediatrics because it has antibacterial potential against a variety of pathogens and is resistant to lactamases. The majority of the research on the pharmacokinetics of these antibiotics in children is based on trials conducted on adults.

It has been observed that during extracorporeal membrane oxygenation, sepsis, and septic shock, pharmacokinetics (distribution volume, clearance) alter, resulting in a drop in the blood level of antibiotics. The amount of antibiotics in the blood is also influenced by the kidneys' functional activity. As a result, this essay is pertinent and produced in the conventional way of original research. The authors used nearly half of the publications published between 2020 and 2025 to support the study's objectives and interpret their own findings.

The study's major goal was to use pharmacokinetic modeling based on children's physiology to establish a method for prescription meropenem that would optimize the antibiotic's dosage while accounting for children's kidneys' filtration capacity. The suggested method of mathematical modeling is appropriate.
Because the study was conducted in a Chinese community with a wide variety of ages and potential genetic and physiological variances, it is doubtful that the results will be 100% replicable. This does not, however, lessen the study's importance.
The study's main conclusions and scope are succinctly summarized in the illustrative material. The data displayed in tables and figures is interpreted by the writers throughout the essay.

The existence of individual variability in the pharmacokinetics of the antibiotic, which is linked to the physiology of the kidneys (filtration capacity) in both normal and pathological conditions, is confirmed by objective data, as is the requirement for an individual choice of both the drug dose and the mode of administration.
The local committee gave its approval and the work was completed in accordance with ethical standards.

Drawbacks:

  1. When an acronym is first mentioned in the text, it should be extended;
  2. The final word of the sentence and the square brackets containing the citation number should be separated by a space;
  3. The phrase "in-vitro" should be written individually and Latin should be italicized; 4. The amount of white blood cells in Table 1 should be stated as 109/L rather than 106/L.

Author Response

Dear Reviewer,

Thank you very much for taking the time to review our manuscript and provide these valuable comments. Your insightful suggestions are essential for improving the quality of our work. We have carefully addressed each of the points you raised, and the corresponding revisions have been made throughout the manuscript. The attachments include: (1) The revised manuscript showing revisions made by authors_ 3966842_R1; (2) The revised manuscript with clean version_3966842_R1_clean; (3) The revised supplementary material showing revisions made by authors_Supplementary Material_R1. All page and line numbers cited in the responses (e.g., Page*, Line*) refer to the revised manuscript file “pharmaceutics-3966842_R1.docx” with tracked changes.

Please find our point-by-point responses below:

1: When an acronym is first mentioned in the text, it should be extended.
Response: We appreciate you pointing out this oversight. We have thoroughly checked the manuscript and have now defined the following acronyms in full upon their first appearance.

e.g.:  PK (Page 3, Line 75); PICU (Page 3, Line 76); PK/PD (Page 3, Line 101); MIC (Page 3, Line 102); ARC (Page 5, Line 194); CL (Page 7, Line 249); IWRES (Page 8, Line 271), and NPDE (Page 8, Line 272).

2: The final word of the sentence and the square brackets containing the citation number should be separated by a space.
Response: We have followed your advice and ensured that a space is inserted between the final word of a sentence and the subsequent citation brackets throughout the entire text.

3: The phrase "in-vitro" should be written individually and Latin should be italicized.
Response: We have corrected this. The term has been standardized to the italicized, stand-alone form "in vitro" accordingly, and we have ensured consistency for similar terms (e.g., in vivo) (Page3, Line97; Page4, Line142).

4: The amount of white blood cells in Table 1 should be stated as 10⁹/L rather than 10⁶/L.
Response: We sincerely thank you for identifying this critical error. We confirm that the unit in Table 1 was incorrect and have now changed it to the standard unit "10⁹/L"(Page6, Tabel1).

Once again, we are grateful for the time and meticulous attention you have dedicated to our manuscript. We believe the manuscript has been significantly improved following your suggestions and hope it now meets the journal's standards.

Reviewer 3 Report

Comments and Suggestions for Authors

The manuscript describes PopPK model-based and PBPK model-based analyses of meropenem dosing in critically ill children. The manuscript is devoted to an important clinical topic. It is well organized and well written. The data analyses and modeling are professional and are of high technical quality. The figures and tables, including the supplementary data, are of high quality and informative. I suggest a limited revision of the manuscript to address the following points of critique:

The incidence of augmented renal clearance in the critically ill pediatric patients is ~12%. Please, present model-based simulations and dose recommendations (Fig. 4, Table 5, an additional graph with the time course of meropenem concentrations) for the 2 populations of critically ill pediatric patients: with and without augmented renal clearance. Please describe the assessment and diagnosis of augmented renal clearance in the individual pediatric critically ill patients, and its implications for meropenem dosing.

Some recent publications on modeling-based optimization of meropenem dosing are not mentioned in the manuscript. Please expand the Discussion part of the manuscript – to compare the results of your analyses (based on PopPK and PBPK modeling) to the data/results/analyses in these and other relevant publications. (Please notice that I am not an author of the references, and is not related to these publications).

Small corrections:

The title of Table 3 – the word “infusion” is missing. Please correct.

Supp Fig. S1 – please rephrase the title, e.g., “local sensitivity analysis, simulation of AUC changes resulting from a twofold increase or decrease in each parameter value”.

English proofing is needed to correct some grammar and punctuation errors, in all the parts of the manuscript.

References:

1. Rapp M, Urien S, Foissac F, Béranger A, Bouazza N, Benaboud S, Bille E, Zheng Y, Gana I, Moulin F, Lesage F, Renolleau S, Tréluyer JM, Hirt D, Oualha M. Population pharmacokinetics of meropenem in critically ill children with different renal functions. Eur J Clin Pharmacol. 2020 Jan;76(1):61-71. doi: 10.1007/s00228-019-02761-7. Epub 2019 Oct 26. PMID: 31654149.
2. Li L, Sassen SDT, Ewoldt TMJ, Abdulla A, Hunfeld NGM, Muller AE, de Winter BCM, Endeman H, Koch BCP. Meropenem Model-Informed Precision Dosing in the Treatment of Critically Ill Patients: Can We Use It? Antibiotics (Basel). 2023 Feb 13;12(2):383. doi: 10.3390/antibiotics12020383. PMID: 36830294; PMCID: PMC9951903.

Comments on the Quality of English Language

English proofing is needed to correct some grammar and punctuation errors, in all the parts of the manuscript.

Author Response

Dear Reviewer,

We sincerely thank you for your positive evaluation of our manuscript and for the insightful comments that have helped us further improve the clarity and depth of our work. We greatly appreciate your recognition of the technical quality of the analyses, the organization of the manuscript, and the relevance of this study to an important clinical topic. We have carefully revised the manuscript according to your suggestions. The attachments include: (1) The revised manuscript showing revisions made by authors_ 3966842_R1; (2) The revised manuscript with clean version_3966842_R1_clean; (3) The revised supplementary material showing revisions made by authors_Supplementary Material_R1. All page and line numbers cited in the responses (e.g., Page*, Line*) refer to the revised manuscript file “pharmaceutics-3966842_R1.docx” with tracked changes. Below, we provide detailed point-by-point responses to all comments.

Regarding the revision needed:

  1. The incidence of augmented renal clearance in the critically ill pediatric patients is ~12%. Please, present model-based simulations and dose recommendations (Fig. 4, Table 5, an additional graph with the time course of meropenem concentrations) for the 2 populations of critically ill pediatric patients: with and without augmented renal clearance. Please describe the assessment and diagnosis of augmented renal clearance in the individual pediatric critically ill patients, and its implications for meropenem dosing.

Response: We sincerely appreciate this insightful suggestion. We fully agree that distinguishing between patients with and without ARC is essential for accurately characterizing pharmacokinetic variability and optimizing meropenem dosing in critically ill children.

  1. a) ARC diagnostic approach

As recommended, we carefully revisited the diagnostic framework for ARC. A major challenge in pediatrics is the lack of a universally accepted criterion analogous to the fixed adult threshold (e.g., CrCl ≥ 130 mL/min/1.73 m²). As highlighted by Smeets et al. (2023), applying adult cutoffs to children fails to account for age-dependent maturation of glomerular filtration and may significantly underestimate ARC prevalence. In alignment with these findings, we adopted a developmentally appropriate, physiology-based definition of ARC: GFR exceeding one standard deviation above the age-specific physiological mean (Mean + 1 SD). This approach reflects both renal maturation and critical illness–related hyperfiltration, and is consistent with recent pediatric GFR reference studies [1–3]. (Page5, Line194-198 and Supplementary Material Page11, Line 12-23).

  1. b) Model-based simulations and concentration–time profiles

Following the reviewer’s recommendation, we generated separate PopPK- and PBPK-based simulations for pediatric patients with and without ARC. Because of the limited availability of high-quality GFR reference data in preterm neonates and to avoid introducing classification uncertainty, the ARC-stratified simulations presented in the main text are restricted to term neonates and older children, for whom reliable physiology-based reference values exist. The corresponding simulations for preterm neonates, as well as all simulated concentration–time profiles from both modeling frameworks, have been added to the Supplementary Materials for completeness and transparency. (Page12, Figure4; Line357-362 and Supplementary Material FigureS4 and S5).

  1. c) Implications for dosing

We further refined dosing recommendations based on the degree to which an individual’s GFR exceeds the age-expected physiological range. These ARC-severity–based recommendations are now provided in the footnotes of relevant tables to facilitate clinical interpretation. (Page13, Table5)

References:

[1] Jančič SG, Močnik M, Marčun Varda N. Glomerular Filtration Rate Assessment in Children. Children (Basel). 2022 Dec 19;9(12):1995. doi: 10.3390/children9121995.

[2] Smeets NJL, Teunissen EMM, van der Velden K, van der Burgh MJP, Linders DE, Teesselink E, Moes DAR, Tøndel C, Ter Heine R, van Heijst A, Schreuder MF, de Wildt SN. Glomerular filtration rate in critically ill neonates and children: creatinine-based estimations versus iohexol-based measurements. Pediatr Nephrol. 2023 Apr;38(4):1087-1097. doi: 10.1007/s00467-022-05651-w.

[3] Smeets, Nori J.L; IntHout, Joanna; van der Burgh, Maurice J.P.; Schwartz, George J.4; Schreuder, Michiel F.; de Wildt, Saskia N.. Maturation of GFR in Term-Born Neonates: An Individual Participant Data Meta-Analysis. JASN 33(7):p 1277-1292, July 2022. DOI: 10.1681/ASN.2021101326 

  1. Some recent publications on modeling-based optimization of meropenem dosing are not mentioned in the manuscript. Please expand the Discussion part of the manuscript – to compare the results of your analyses (based on PopPK and PBPK modeling) to the data/results/analyses in these and other relevant publications. (Please notice that I am not an author of the references, and is not related to these publications).

Response: We sincerely thank the reviewer for this valuable suggestion. As recommended, we have expanded the Discussion section to compare our findings with those of other relevant and recent publications. The revised text now reads:

Across six commonly used clinical scenarios, both models produced similar central estimates of steady-state exposure but converged in identifying a critical limitation: standard regimens are frequently insufficient, and shortfalls worsen with higher MIC and enhanced renal function (high eGFR/ARC). In line with this, the PopPK study by Rapp et al. reported a median eGFR of 151.5 mL/min/1.73 m² in their cohort, with widespread ARC identified as the primary driver of accelerated MEM clearance and failure to achieve target plasma concentrations [37]. When MIC is low and renal function is normal, model-based doses align with label/guideline recommendations; however, in ARC-prone patients or at higher MICs, prolonged infusion and/or increased dosing frequency are warranted. Simulations from both models revealed substantial interindividual variability in MEM pharmacokinetics and PTA, even within narrow eGFR ranges. This variability reflects dynamic changes in renal function and other treatment-related factors , such as fluid resuscitation during severe infection, which lead to substantial interindividual differences in drug exposure. These findings underscore the potential utility of TDM, particularly in unstable patients, to ensure optimal exposure. This interpretation is further supported by the external evaluation of Li et al., which demonstrated that existing MEM models perform poorly in a priori prediction but predictive accuracy improves markedly with the integration of just one or two TDM samples [38]. Based on these findings, we advocate for a comprehensive clinical management pathway: initial therapy should employ optimized strategies such as prolonged or continuous infusion for patients with ARC or high-MIC pathogens , followed by early implementation of TDM within a validated PK model framework to enable fully individualized therapy. (Page14, Line422-453)

  1. Rapp M, Urien S, Foissac F, Béranger A, Bouazza N, Benaboud S, Bille E, Zheng Y, Gana I, Moulin F, Lesage F, Renolleau S, Tréluyer JM, Hirt D, Oualha M. Population pharmacokinetics of meropenem in critically ill children with different renal functions. Eur J Clin Pharmacol. 2020 Jan;76(1):61-71. doi: 10.1007/s00228-019-02761-7. Epub 2019 Oct 26. PMID: 31654149.
    38. Li L, Sassen SDT, Ewoldt TMJ, Abdulla A, Hunfeld NGM, Muller AE, de Winter BCM, Endeman H, Koch BCP. Meropenem Model-Informed Precision Dosing in the Treatment of Critically Ill Patients: Can We Use It? Antibiotics (Basel). 2023 Feb 13;12(2):383. doi: 10.3390/antibiotics12020383. PMID: 36830294; PMCID: PMC9951903.

Regarding the specific minor corrections raised:

  1. The title of Table 3 – the word “infusion” is missing. Please correct.

Response: We thank the reviewer for pointing out the omission. The title of Table 3 has been corrected to include the word "infusion."(Page 9, Line 305)

  1. Supp Fig. S1 – please rephrase the title, e.g., “local sensitivity analysis, simulation of AUC changes resulting from a twofold increase or decrease in each parameter value”.

Response: We agree with the suggestion for greater clarity. The title of Supplementary Figure S1 has been rephrased as suggested, to: "Local sensitivity analysis: simulation of AUC changes resulting from a twofold increase or decrease in each parameter value." (Supplementary Material Figure S1)

  1. English proofing is needed to correct some grammar and punctuation errors, in all the parts of the manuscript.

Response: We thank the reviewer for highlighting the need for language polishing. In response, the entire manuscript has undergone thorough professional English editing by a qualified editing service to rectify grammatical and punctuation errors, and to enhance overall clarity and readability. The revised text has been tracked in the " pharmaceutics-3966842_R1.docx" file using the "Review" function.

Once again, we thank the reviewer for the constructive comments and the careful evaluation of our manuscript. We believe that the revisions substantially strengthen the scientific rigor and clinical relevance of our study.
